# Statistical modeling of SARS-CoV-2 substitution processes: predicting the next variant

Keren Levinstein Hallak[1] & Saharon Rosset [1✉]

We build statistical models to describe the substitution process in the SARS-CoV-2 as a function of explanatory factors describing the sequence, its function, and more. These models serve two different purposes: first, to gain knowledge about the evolutionary biology of the virus; and second, to predict future mutations in the virus, in particular, non-synonymous amino acid substitutions creating new variants. We use tens of thousands of publicly available SARS-CoV-2 sequences and consider tens of thousands of candidate models. Through a careful validation process, we confirm that our chosen models are indeed able to predict new amino acid substitutions: candidates ranked high by our model are eight times more likely to occur than random amino acid changes. We also show that named variants were highly ranked by our models before their appearance, emphasizing the value of our models for identifying likely variants and potentially utilizing this knowledge in vaccine design and other aspects of the ongoing battle against COVID-19.

[1] Department of Statistics and Operations Research, School of Mathematical Sciences, Tel-Aviv University, 6997801 Tel-Aviv, Israel.
✉email: saharon@tauex.tau.ac.il

The intense community effort of SARS-CoV-2 sequencing has yielded a wealth of information about the mutations that have occurred in the virus since it first appeared in humans.

Understanding the evolutionary dynamics of the virus is critical for inferring its origin[1,2], understanding its underlying biological mechanisms like mutagenic immune system responses[3,4] and recombination[5,6], predicting virus variants[7–9], and for vaccine and drug development[10,11]. Recently there has been a spur of interest in analyzing substitution rates for SARS-CoV-2[12–14]. Common analyses relate to explaining factors such as genes[15–17], CpG pairs[18,19], context[13,20], and codon and amino acid frequency[21,22]. However, all previous work relied on a statistical analysis of the effect of each factor in isolation through summary statistics. If we seek to gain a deeper understanding and utility, we should consider these factors in tandem and aspire to build models that describe the entire mutation process as a function of all relevant information. In addition, while phylogenetic methods have been useful for finding and categorizing current variants, they have not been used for predicting new variants.

In this work, we employ regression in a big-data approach to identify the best statistical models for explaining the substitution rate distribution in observed sequences. We build a dataset containing 51,527 inferred substitutions for training the models based on a phylogenetic tree reconstruction from 61,835 available sequences[23] (as of 8 February 2021). We use the inferred substitutions in these sequences to identify the factors affecting substitution rates at different locations in the viral genome. We use our learned model to predict which sites in the genome are likely to mutate in the future and contribute to the formation of novel variants. Our methods can help vaccine design, medical research, and other tasks in the ongoing battle against COVID-19 and future viral epidemics.

We consider two different candidate phylogenetic trees: Tree of complete SARS-CoV-2 sequences reconstructed by NCBI[23] and a phylogenetic tree we reconstructed by applying the sarscov2phylo method developed by Lanfear[24] on the same sequences. Here we show results on the latter; we provide results for the NCBI phylogenetic tree in the Supplementary Information.

In our models, we consider ten potential explanatory factors for explaining substitution rates based on sequence, biological function, gene location, and others. We compare 43,254 possible regression models and choose between them based on statistical goodness of fit scores.

We evaluate the ability of these models to predict new variants appearing in sequences that were added to the NCBI database between 10 February 2021 and 10 April 2021 (the test period). Our evaluation scheme does not depend on the correctness of the inferred tree or the family of regression models, thus objectively evaluating our models' ability to rank potential variants. For example, while the overall rate of occurrence of new amino acid substitutions in the test period was 2.2% among all candidate sites, the top 100 predictions of our selected model included 19 substitutions that actually occurred in the test period, for a lift (excess precision compared to random ranking) of 8.62.

## Results

**SARS-CoV-2 substitution model**. We briefly describe our statistical modeling approach here; See the "Methods" section for more details.

We inferred a phylogenetic tree and its mutations from the 44,080 sequences that passed quality control (out of the 61,835 sequences available in the NCBI dataset as of 2/8/2021). We then built a training dataset describing all potential substitutions in terms of the following explanatory factors:

1. Locus (Gene) of the site considered
2. Input nucleotide base (A/C/G/U)
3. Input amino acid
4. Input codon
5. The position of the site in the codon (1–3)
6. Mature peptide indicator
7. Stem loop indicator (different categorical values for each one of the stem loop genes ORF10 and ORF1ab)
8. CG pair indicator (different value for each position of the CG pair or NULL for non-CG)
9. Right neighboring nucleotide
10. Left neighboring nucleotide

We considered all possible combinations of using each factor in a generalized linear model (GLM)[25]: (—) omission, (+) as an explanatory factor, or (/) using it to split the GLM into sub-models such that a separate sub-model is built for each possible value. In our nomenclature, a model denotes a specific choice of inclusion (—,+,/) for each one of the categorical factors, and we fit the data the sub-models created by splitting according to the (/) factors. Subsequently, a total of 43,254 models were examined (each comprised of multiple sub-models). To account for over-dispersion, we considered a Negative-Binomial (NB) regression model in addition to the standard Poisson regression model in our GLM. All our models were fitted separately to synonymous and non-synonymous substitutions and accounted for the difference in rates between transitions and transversions.

Figure 1 shows the top three NB and Poisson regression models based on their AIC (penalized log-likelihood) score[26] on the training dataset. Please refer to the Supplementary Information for similar analyses of the NCBI phylogenetic tree (Supplementary Note 1 and Fig. S1) and ten top models (Fig. S2). In addition, we provide all models in the files Supplementary Data 1 and Supplementary Data 2.

**Predictions**. We next evaluated the ability of our top models to predict novel substitutions. Our prediction data set was constructed as follows. We considered the 32,495 test sequences that were added to the NCBI database in the period between 10 February 2021 and 10 April 2021. We then identified 10,409 sites with zero substitutions in the training data, i.e., identical or missing in all training data sequences. Of which 9696 sites had at

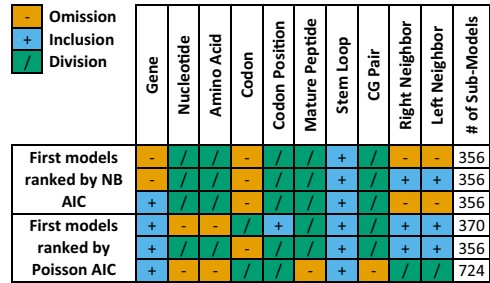

**Fig. 1 Top-scoring models for the training dataset.** The first three rows correspond to the top-scoring models when NB regression is applied. The next three rows correspond to the top-scoring models when Poisson regression is used. Each explaining factor is either (—) omitted from the model, (+) used as an explanatory factor, or (/) used to split the GLM into sub-models. We note that there are potential redundancies in the models. For example, the codon explaining factor contains the complete information on the amino acid and the nucleotide explaining factors (but not the other way). Our regression method of examining all inclusion possibilities for each factor considers this and produces a precise score regardless of the intertwined information.

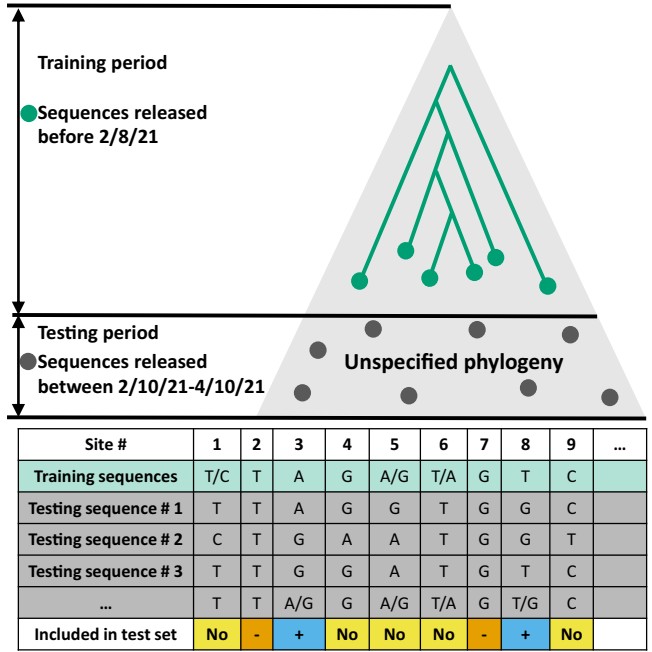

| Site # | 1 | 2 | 3 | 4 | 5 | 6 | 7 | 8 | 9 | ... |
|---|---|---|---|---|---|---|---|---|---|---|
| Training sequences | T/C | T | A | G | A/G | T/A | G | T | C | |
| Testing sequence # 1 | T | T | A | G | G | T | G | G | C | |
| Testing sequence # 2 | C | T | G | A | A | T | G | G | T | |
| Testing sequence # 3 | T | T | G | G | A | T | G | T | C | |
| ... | T | T | A/G | G | A/G | T/A | G | T/G | C | |
| Included in test set | No | - | + | No | No | No | - | + | No | |

**Fig. 2 An illustration of the training and testing dataset for prediction.** Our training data consists of a phylogenetic tree reconstruction based on sequences released before February 8th, 2021 (green dots). The test data is comprised of sequences that were released between February 10th and April 10th, 2021 (gray dots). For these, we did not infer a phylogeny or rely on any other phylogenetic information. To evaluate our ability to predict new substitutions, we considered only sites for which no substitutions had occurred in the training data. The table in the figure shows examples of which substitutions are included in the test dataset. For sites 1, 5, and 6, the base is not constant for the training data set, and therefore it is not included in the test dataset. In sites 4 and 9, there is only one sequence in the test set that shows a different base from the training sequences; these sites have not been included in the test set to avoid sequencing errors. For sites 2 and 7, the base is constant for both the training and the test dataset making them negative examples in the test dataset, whereas sites 3 and 8 are positive examples, where a confirmed substitution occurred in the test period.

most one base different from the base appearing in the training data, allowing us to confidently identify the substitution that occurred without inferring a phylogenetic tree for the test sequences. In these, we identified 2697 sites that had at least one substitution in the test sequences. To avoid labeling sequencing errors, we required a minimum of two different test sequences with the mutated state; hence only 1266 sites remained. Sites that had a single test sample with a mutated state were entirely ignored in the evaluation phase. For an illustration of the training and test datasets and our labeling procedure, see Fig. 2.

We evaluated the ability of the top regression models to successfully rank the sites by their likelihood to mutate during the test period, thus creating new variants. Our evaluation is done at the amino acid level rather than the individual site (nucleotide) level to express the notion that non-synonymous amino acid changes are the true object of interest in predicting new variants. The transition from predicting sites to predicting amino acids is done by careful post-processing and aggregation of the prediction model results (see the "Methods" section). We used the area under the ROC curve (AUC) and the lift (ratio of true positives compared to a baseline model) to assess our results. The lift compares our model to two baselines: the random ordering of all

possible relevant substitutions and a base model, which takes into account *exposure*, i.e., the number of ways in which a specific amino acid can be created, and also the transition/transversion (ti/tv) ratio, but not the other explanatory factors. We compared to the base model as a sanity check that our models were indeed finding additional information to characterize amino acid substitution rates beyond the exposure and ti/tv effect.

The results for our top models are shown in Fig. 3, both for the entire viral genome and the spike gene only, due to its biological importance[27]. We use both Poisson and Negative Binomial regressions to predict the substitution rate for each model. Synonymous and non-synonymous substitution rates are modeled separately due to the fundamentally different biological and evolutionary mechanisms they trigger. The community interest in non-synonymous substitutions also supports this separation[28,29]. Note that the substitutions are aggregated per amino acid and location on the genome, as explained in the "Methods" section. Figures S3 and S4 show the results for NCBI's phylogenetic tree and our top ten models.

Based on these results, we chose the third Poisson model of non-synonymous amino acid substitutions for a more detailed presentation here. The lift curves for this model are shown in Fig. 4, demonstrating in more detail our models' ability to identify likely substitutions. Note that in the test dataset, there are roughly 2% positives. Using the calculated lifts at 1%, the number of true positives is 7.51 times greater than the random model and 3.125 times greater than the base model. In numbers, this 1% represents 337 candidate substitutions, of which 50 actually occurred in the test period (compared to 6.66 expected under the random model and 16 in the top base model predictions). The lift curve against the base model is lower than that against the random model, yet still much higher than 1 for the highly ranked candidates (left side of the plot). This demonstrates that the exposure information used in the base model is essential for successful prediction, but the detailed models can still identify a substantial signal beyond the exposure. Figure S5 shows similar results for NCBI's phylogenetic tree (source data: supplementary data 4).

In order to further validate our model, we have used an additional test set that contains sequences collected between 15 September 2021 and 1 October 2021. This test set also produces similar results to the one presented here (see Supplementary Note 2 and Figs. S6 and S7, source data for Fig. S7 appears in supplementary data 5).

To help the community predict and analyze future substitutions, we provide a complete list of predicted non-synonymous amino acid substitution rates in the spike protein in the file Supplementary Data 6. In addition, we note for each substitution whether or not it was observed in the training and test datasets.

As an additional demonstration of our models' success in ranking amino acid substitutions of interest, we analyzed the following variants: Alpha (lineage B.1.1.7), Beta (lineage B.1.351), Gamma (lineage P.1), Delta (lineage B.1.617.2), Theta (lineage P.3), Omicron (lineage B.1.1.529), Lambda (lineage C.37), Mu (lineage B.1.621), Epsilon (lineages B.1.429, B.1.427), Zeta (lineage P.2), Eta (lineage B.1.525) and Theta (lineage P.3). Many of the amino acid substitutions are common to several variants. Overall, there are 72 different amino acid substitutions in the spike protein comprising these variants. Of these, 45 were included in the training data, while 27 were recorded after our training cutoff date of 2/8/2021. According to our chosen model (third-ranked Poisson model), we examined their ranking in the 13,544 possible spike protein amino acid substitutions list. A list of all 72 amino acid substitutions and their rankings is given in Fig. S8, demonstrating that 68% of the substitutions (49/72, including 16 substitutions not observed in training) were ranked in the top 2735 predictions (that is, top 20% of predictions)

|  | Model # | Non-synonymous amino acid substitutions | | | | | | Synonymous amino acid substitutions | | | | | |
|---|---|---|---|---|---|---|---|---|---|---|---|---|---|
|  |  | Poisson | | | Negative Binomial | | | Poisson | | | Negative Binomial | | |
|  |  | AUC | 3% Lift Vs. | | AUC | 3% Lift Vs. | | AUC | 3% Lift Vs. | | AUC | 3% Lift Vs. | |
|  |  |  | Random model | Base model |  | Random model | Base model |  | Random model | Base model |  | Random model | Base model |
| All genes | 1 | 0.835 | 4.707 | 2.238 | 0.821 | 4.607 | 1.957 | 0.858 | 3.577 | 1.465 | 0.856 | 3.577 | 1.432 |
|  | 2 | 0.832 | 4.406 | 2.095 | 0.819 | 4.306 | 1.830 | 0.861 | 3.861 | 1.581 | 0.858 | 3.463 | 1.386 |
|  | 3 | 0.836 | 5.358 | 2.548 | 0.826 | 4.557 | 1.936 | 0.847 | 3.520 | 1.442 | 0.846 | 3.690 | 1.477 |
| Spike gene | 1 | 0.814 | 4.062 | 2.667 | 0.786 | 2.538 | 1.250 | 0.867 | 4.748 | 3.333 | 0.861 | 1.899 | 1.333 |
|  | 2 | 0.814 | 4.062 | 2.667 | 0.781 | 3.554 | 1.750 | 0.864 | 4.273 | 3.000 | 0.859 | 3.798 | 2.667 |
|  | 3 | 0.830 | 4.062 | 2.667 | 0.827 | 3.554 | 1.750 | 0.863 | 4.748 | 3.333 | 0.864 | 4.748 | 3.333 |

**Fig. 3 Prediction results for the top three models.** We use the top three Poisson and Negative Binomial models from Fig. 1 for prediction on the test dataset. Results for the entire genome are in the first three rows, for the spike protein only in the last three. Results are shown separately for predicting non-synonymous amino acid substitutions (left half) and predicting synonymous substitutions (right half, these results are not discussed in the text). The first column in each table quarter shows the area under the ROC curve (AUC) for the corresponding prediction task and modeling approach. We highlighted the top-scoring model for every (substitution type, locus, approach) combination. Overall, we obtained high AUC scores, showing that the models successfully predicted many of the substitutions. Each quarter's second and third columns are 3% lift scores of each model versus the random and more elaborate base models (see text and Methods). The top models significantly outperform both baselines, stressing our approach's benefits over more naive statistical predictions. The model we analyzed further in the text (third Poisson model for non-synonymous amino acid substitutions) is also red-framed.

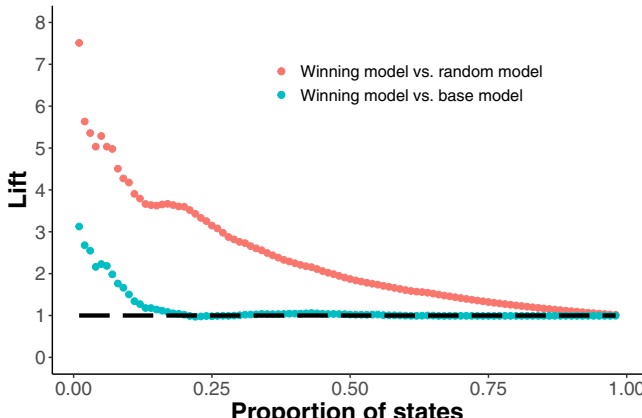

**Fig. 4 Lift curves of the winning model versus the random (red) and base models (cyan).** We compare the winning model, the third-ranked Poisson model of non-synonymous amino acid substitutions, against two baselines. The first is the random ordering of all possible relevant substitutions (red), and the second is a base model, which considers the exposure and the transition/transversion (ti/tv) ratio, but not the other explanatory factors (cyan). To compare, we show the lift score (the ratio of true positives compared to a baseline model) as a function of the proportion of states considered. Source data: supplementary data 3.

according to our model. In Fig. 5 we provide a similar analysis separately for the latest Omicron variant, showing that 70% of its spike protein amino acid substitutions (21/30, including 11 substitutions not observed in training) were ranked in the top 2733 predictions (that is, top 20% of predictions) according to our model. This result is also very significant ($p < 2.2e - 16$, one-sided Wilcoxon rank-sum test, test statistic: $W = 777,362$, 95% CI: (3476, ∞)).

Some of the substitutions comprising the inspected variants are hypothesized to be the result of positive selection[30–33]. As our model does not take positive selection into account, we would expect them to be ranked as less likely to occur by our model, compared to the non-selected mutations. In order to test this hypothesis, we conducted a one-sided Wilcoxon rank-sum test of whether substitutions having a survival advantage come from the same distribution as the rest of the 72 substitutions comprising the inspected variant. We identified a list of mutations noted in

the literature as potentially conferring a selective advantage: S477G/N[34], E484Q[35], N501Y[36], N501S[37] enhancing binding of the spike protein to the hACE2 receptor; L452R[38], N440K[39], D614G[40] conferring increased infectivity; G446V[41], E484K[42] affecting the affinity of monoclonal antibodies; and F490S[41] reducing susceptibility to an antibody generated by those who were infected with other strains. Our test rejects the null hypothesis that this sub-group of substitutions comes from the same distribution as the rest of the 72 substitutions ($p = 0.0066$, test statistic: $W = 106,589$, 95% CI: (987, ∞)). This observation suggests that beyond these identified mutations, other high prevalence substitutions with a low probability of mutation in our models may also be under positive selection.

## Discussion

In this work, we model substitution rates in the SARS-CoV-2 as a function of several possible affecting factors describing sequence and coding information. We fit our models to training data that is based on inferring the phylogenetic tree connecting tens of thousands of sequences collected before February 2021 and also inferring the specific substitutions that have occurred on this tree. This phylogenetic reconstruction task is extremely challenging, and it is unlikely that the inferred tree or substitutions are completely accurate[14]. This is also evident by the different trees, substitutions, and slightly different models we get when we use the sarscov2phylo method[24] to reconstruct the tree, with results given in the main text, compared to using NCBI's reconstruction of the tree (see Supplementary Note 1 and Figs. S1, S3, and S5).

However, a critical point is that our evaluation approach on the *test set* of sequences added after the training cutoff date does not rely on any phylogenetic reconstruction or assumptions on the phylogenetic context between the test sequences and training sequences (as illustrated in Fig. 2). The fact that the test set shows high AUC and lift curves demonstrates that regardless of doubts about the accuracy of the training phylogenetic reconstruction, the models we fit to the training data are indeed useful to predict future substitutions.

The specific substitutions we include in the test set were carefully chosen to avoid sequencing errors and phylogenetic uncertainty in the evaluation. However, we emphasize that our models can be used to predict the likelihood of all possible substitutions and variants, including ones that have already appeared

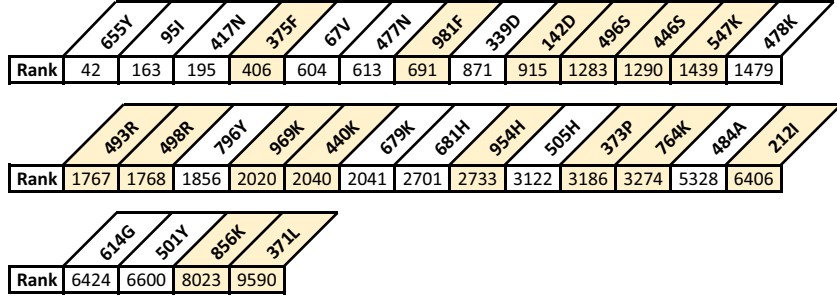

**Fig. 5 Rank of spike protein amino acid substitutions.** Ranking was performed by our prediction model on 13,544 possible non-synonymous amino acid substitutions in the spike protein resulting from one nucleotide change. The ranks of the 30 substitutions comprising the Omicron variant (lineage B.1.1.529) are shown. The highlighted substitutions were not part of the training dataset.

in the training data (as we did in our analysis of known variants in Fig. 5). Furthermore, the nucleotide level predictions we generate can be easily transformed into amino acid level predictions, as we did in our actual evaluation and AUC and lift calculations (with the methodology described in the "Methods" section). This is critical since the discussion of variants in the literature is typically focused on the amino acid level[43,44].

Our top regression models shown in Fig. 1 suggest that all of the factors we consider are potentially useful for predicting future substitutions and variants, but some are more important than others. Specifically, most of the best models split into sub-models by amino acid rather than by codon (as shown by their designation as '/' in all top models according to NB AIC), suggesting that codon usage bias effects such as those described in refs. [17,28] may not be major.

An important property of our regression approach is that regression models consider all candidate explanatory factors at once. They are thus able to identify factors that appear essential when considered on their own but whose effect can be explained away by other, better factors. For instance, the neighboring nucleotides identities (context) seem to have a minor role once the amino acid and codon position are taken into account and are not included at all in some of our top models (as indicated by their designation as—in two of the top three models). While it is true that in an analysis examining only the connection between neighbors and likelihood of substitution, the context would appear very significant, this effect is mitigated and may disappear when taking into account the better factors (see model 21,532 in Supplementary Data 1).

Our analysis includes ten variables that can affect the substitution rate. Many others can be proposed, including sequence-based variables such as more elaborate sequence contexts than immediate neighbors and external information such as conservation scores. As more data and knowledge accumulate, we expect our prediction models to improve by adding such relevant variables.

In summary, our statistical modeling approach offers two substantial benefits: A better understanding and modeling of the factors affecting substitution rates in the SARS-CoV-2 virus, and by implication in other viruses, and the resulting predictive models, which can be used to rank future variants by their likelihood.

Our contributions can potentially play a role in vaccine design, medical research, and other tasks in the ongoing battle against COVID-19 and future viral epidemics. Specifically, for the important task of vaccine design, one can imagine future pipelines where vaccines for many different potential variants can be prepared in advance using mRNA technology. Prioritizing which potential variants are more relevant can be done based on a combination of mutation likelihood prediction tools like we offer,

with tools for inferring other relevant aspects like infectiousness[9] and target effectiveness[45]. In addition, we demonstrated that high prevalence substitutions that hold a survival advantage are typically not identified by our models as having a high mutation rate. This observation suggests our models can be used to flag additional candidates for study as potentially inducing positive selection.

## Methods

**Statistics and reproducibility.** The sequences used in this work were all downloaded from the NCBI website[23,46]. As a training set, we used 61,835 available sequences as of 8 February 2021. For a test set, we used 32,495 sequences released between 10 February 2021 and 10 April 2021. NCBI's tree[47] and the sarscov2phylo method[24] exclude noisy sequences. These include low-quality sequences and sequences missing sufficient data, so that, it is hard to place them meaningfully in the phylogeny. In addition to the results presented in the main text for the phylogenetic tree reconstructed according to the sarscov2phylo method, we provide results for the NCBI phylogenetic tree showing similar results (see Supplementary Note 1, Figs. S1, S3, and S5). To further validate our model, we have used an additional test set that contains sequences collected between 15 September 2021, and 1 October 2021. This test set also produces similar results to the one presented in the main text (see Supplementary Note 2 and Figs. S6 and S7). The threshold date separating the training and test sequences was chosen once and arbitrarily. To reproduce the results, the code used in this work is available at[48].

**Phylogeny of SARS-CoV-2.** We used two phylogenetic reconstructions of SARS-CoV-2 following related works in the literature[13,49,50]:

1. The tree of complete SARS-CoV-2 Sequences by NCBI[47]. This is a distance-based phylogenetic tree. Further information is available online[51,52].
2. A tree reconstructed by us using the sarscov2phylo method developed by Lanfear[24]. Following Lanfear's method, we estimated the global phylogeny using IQ-TREE2[53] and FastTree 2[54]. The resulting tree was then rooted with the NCBI reference sequence (accession *NC_045512.2*) using nw_reroot[55]. Finally, we removed sequences from very long branches using TreeShrink[56].

We used the global sequence alignment method implemented in the sarscov2phylo method which aligns every sequence to the reference sequence (accession *NC_045512.2*) from NCBI and then joins the individually aligned sequences into a global alignment using MAFFT v7.471[57], faSplit[58], faSomeRecords[59], and GNUparallel[60].

**Internal nodes reconstruction.** The internal nodes of the tree phylogeny are necessary to infer the substitutions that occurred on the tree edges. We now describe our heuristic, inspired by Fitch's algorithm[61], used to reconstruct the sequences in the internal nodes. Model-based approaches for ancestral sequence reconstruction (such as FastML[62]) cannot be applied here due to a large number of sequences.

Every site holds a probability vector over the bases A/C/G/U defined as follows:

1. For every leaf, assign probability 1 to the base in the respective site and probability 0 to all other bases. The probability is split uniformly among the possible bases whenever there is base ambiguity.
2. Pass from bottom to top. The probability vector of an internal node is the average of the probability vectors of its children.
3. Pass from top to bottom. We descend the tree from the root and add to each node $\epsilon = 1/(\# \text{ of children})$ multiplied by its parent's probability vector (and normalize by $1 + \epsilon$ to keep it in the $l_1$-simplex).
4. The chosen base at every node is determined by the highest probability value. This procedure also solves ambiguous sites in the leaves.

By doing this, we break ties between the highest probabilities (such ties are frequent) and allow information to flow between nodes that have a common ancestor.

Finally, we applied a battery of statistical tests to validate the phylogenetic tree and its internal nodes (details in Supplementary Note 3).

**Substitution model**. By reconstructing the tree's internal nodes, we can generate a tabular dataset consisting of the list of factors and the number of substitutions that occurred for each instantiation of these factors. We use the multiple regression approach described in ref. [25] which considers for every factor in the tabular data the options to either join in the regression linearly (marked '$+$'), not join at all (marked '$-$'), or to partition the data according to it (marked '$/$'). We use the term model to denote a specific choice of inclusion for each categorical factor that might affect the substitution rate as listed.

A partitioning ($/$) splits the regression model into multiple smaller regressions, where each factor gets one of its values. Consider, for example, that there are only two factors, the base and the codon position. If both are ($+$), then only one regression will be applied with a one-hot encoding of both factors. However, if the base is ($/$), we will use four regression models to partition the data according to the base (A/C/G/U). We use the term sub-model for each of the actual models fitted after splitting. The AIC[26] score is given by $AIC = 2k - 2\log(\hat{L})$ where $k$ is the number of free parameters and $\hat{L}$ is the maximum likelihood. The AIC score is calculated separately for each sub-model regression. Then, the AIC scores of these sub-models are summed up to form one unified score for this model.

Consequently, the number of models we consider is, in theory, combinatorial in the number of values each factor can have. However, the number of models can be substantially reduced since some factors are dependent on one another (for example, the codon determines the amino acid and base). In our data, we score 43,254 models. We apply both Poisson regression and Negative-Binomial regression[63] for each model, where the latter is used to account for overdispersion, specifically to account for latent factors not included in the model. The complete list of factors is given in the main paper. Finally, our experiments infer different regression coefficients for synonymous and non-synonymous sub-models and combine the AIC scores. We also considered doing the same for transitions/transversions and different output nucleotides, but we got strictly worse AIC scores.

Another critical notion is that of exposure[64], which weights the states we train on according to the frequency of their occurrence. For instance, a specific combination of frequently appearing factors in the dataset has relatively higher exposure than a rare set. When we learn the regression model, taking exposure into account is crucial to reduce bias in the dataset and improve the predictions. The exposure is proportional to the total amount of time a specific set of factors was observed. To calculate that duration, we summarize the lengths of relevant branches in the phylogenetic tree and use the sum as an offset variable in the regression. For the test set, exposure is unnecessary (or can be set to an arbitrary constant) as we calculate the exposure for the leaves of the tree, which are the training sequences, and we only consider sites for which there were no substitutions along the phylogenetic tree.

Finally, we apply additional normalization. We first define the non-synonymous ti/tv ratio[65]:

$$r_{ti:tv}^{non-syn} = \frac{\#Non-synonymous\ transitions}{\#Non-synonymous\ transversions}$$

in the training data. Then, we count the number of possible transitions and transversions per state for each state and normalize the substitution rate accordingly. For example, the codon GCG has one possible non-synonymous transition and two possible non-synonymous transversions in the first codon position. The non-synonymous substitution rate for that state is hence normalized by $1 + 2/r_{ti:tv}^{non-syn}$. An identical procedure is applied to the synonymous substitutions.

**Prediction**. Our main prediction task is focused on predicting amino acid substitutions. As our basic predictions are always at the single nucleotide level, we carefully aggregate them to form amino acid predictions—the substitution rate of an amino acid output at a given location is the sum of the rates of all the substitutions leading to it. Note that in most but not all cases, there is only a simple correspondence, in that there is a single non-synonymous nucleotide substitution that leads to a given amino acid change. However, more complex settings can occur, such as the substitution from Histidine to Glutamine through four different non-synonymous transversions in the third codon position.

To test the performance of our predictions, we compare them to two baselines. The first baseline is the random model which places equal probability on all amino acid substitutions. While a naive random model would consider all 21 amino acids per location, we permit only one substitution per codon since multiple substitutions per codon are highly unlikely (<0.5% of the substitutions occurred at adjacent sites in the same tree branch). This limitation drastically improves the random model's predictions and reduces possible amino acid substitutions throughout the molecule from 121,653 to 33,684.

The second baseline model is called base model. This model considers the exposure and ti/tv normalization for each substitution and uses it for prediction. Hence it is a lot less naive than the random model and relies on careful evaluation of the different likelihoods for different substitutions based on the observed states in the tree and the ti/tv effect. It differs from our true prediction models in ignoring the ten potential affecting factors, and comparing to it is our way to quantify the contribution of these factors to predictive power within our regression approach.

To compare the top models to the baseline models, we use two scoring methods—AUC and lift (we emphasize here again that all comparisons are made on data in the test period not used for building the models, as explained in Fig. 2 of the main text). First, we transform the predicted substitution rate into a binary prediction vector of 0/1 predictions. We do this by applying a threshold on the predicted substitution rate where all rates above a specific value are deemed positive. By varying the threshold, we can derive the ROC curve (using the test dataset as the ground truth), from which we can calculate the AUC score. Lift[66,67] measures how well a targeting model performs at predicting compared to a random choice method. We compute the lift for each threshold by taking the ratio of "precision at x%" between our model and each baseline model separately.

**Reporting summary**. Further information on research design is available in the Nature Research Reporting Summary linked to this article.

## Data availability
The datasets generated during and/or analysed during the current study are available from the corresponding author on reasonable request.

## Code availability
The code used in this work is available at: https://github.com/Kerenlh/sarscov2predictions/releases/tag/1.0.0, https://doi.org/10.5281/zenodo.5831603.

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

## Acknowledgements

This work was partially supported by Israeli Science Foundation grant 2180/20. We thank the Edmond J. Safra Center for Bioinformatics at Tel-Aviv University for a fellowship partly supporting this work.

## Author contributions

S.R. supervised research, K.L.H. performed the experiments and analyzed the data, S.R. and K.L.H. conceived and designed the experiments, performed statistical analysis and wrote the paper.

## Competing interests

The authors declare no competing interests.
