## [Peer Review File · Communications Biology]

Reviewers' comments:

Reviewer #1 (Remarks to the Author):

In this work, the authors applied generalized linear model to model and predict nucleotide mutations and amino acid substitutions for SARS-CoV-2 by taking into account explanatory factors related to sequence context, genes, amino acids, etc. They trained models by exhausting all possible combinations of relevant factors on a large NCBI dataset and evaluated the top models with an independent testing dataset also collected from NCBI at a later time period. They showed that highly ranked candidates of amino acid substitutions picked up by the top models are more likely to appear than random amino acid changes, and that variants of interest are highly ranked by their models. They also showed that their approach is robust with respect to phylogenetic tree reconstruction methods. In general, this is an important topic during the ongoing battle against COVID-19, given that the pandemic is still not behind us and new variants of the virus keep emerging. Nevertheless, I have a few concerns and questions about the manuscript.

Major comments:

1) Some of the significances of the study may be overstated. For example, the authors said "Our methods can help vaccine design" in the Introduction (and similar statement in the Abstract), but the path to apply the methods and results from this study to vaccine design is still unclear to me -- although some of the variants of interest are highly ranked (Table 3), there are still thousands of other candidates at similar ranks and it is challenging to pick up the few most important ones from them, and prediction of the prevalence of the variants in the population is a different topic.

2) The authors said the testing data did not depend on the construction of the phylogenetic tree (page 11 lines 184 - 187), while the exposure was calculated based on the phylogenetic tree (page 15 lines 315 - 318). So how was the exposure determined for testing data? Did it implicitly depend on the constructed phylogenetic tree? As the SARS-CoV-2 virus is dynamically evolving, a relevant question is how the approach would perform for the new sequences and new amino acid changes emerging after April 10, 2021, further apart from the period during which the testing data was retrieved for the study (i.e. February 10, 2021 to April 10, 2021, which is immediately after the training cutoff date)?

3) How would the proposed approach perform as compared to existing methods? The authors said previous methods considered each predictor in isolation (page 2 line 34), so it may be informative to compare the top models with the base model + single factor one at a time in terms of AUC and lift.

4) What other sequence context relevant "internal" factors and other "external" factors beyond sequence relevant information could be possibly informative to predict variants of interest? This may be discussed in the Discussion section for future improvement of the prediction model as more data are accumulated.

Minor comments:

1) On page 13, lines 272-273: "Finally, we applied a battery of statistical tests to validate the phylogenetic tree and its internal nodes." Although the authors gave an example, what are these statistical tests? Perhaps more details can be provided in supplementary information.

2) References 24 and 25 are the same.

Reviewer #2 (Remarks to the Author):

This manuscript describes a new method for reporting a list of potential mutations on SARS-CoV-2 genomes that might lead to a new future variant. The model was trained of past data (before ~Feb 2021). It was then used to validate the accuracy for new data (from Feb 2021) and shows

promising results. This topic is highly relevant and might gain a high interest in the scientific community. However, I have some concerns mainly about the methods that I believe need to be addressed. My comments are listed below:

1. The ms didn't make clear one selling point, that traditional phylogenetic methods can't predict future variants, although they're very good at finding and classifying existing variants based on the current data. This can be mentioned in the abstract and/or introduction.
2. Is there a reason why you only used the top-3 models but not top-10 or so?
3. Page 9, Figure 2 caption: Do you mean "random (red) and base models (cyan)"? (colors swapped).
4. Page 12, code availability, this is my main concern: The GitHub repo contains a collection of R scripts but absolutely no guide how to use them. Eventually I can't do any analysis such as replicating the analysis done here. Readers will have the same problem if this is published. So please describe step-by-step usages of your scripts for reproducibility. Moreover, someone may want to use it to re-train the model on new data.
5. Page 12, Phylogeny of SARS-CoV-2: the authors used two unpublished methods from NCBI and sarscov2phylo github repo. That's OK but you need to explain these two methods in more details. For example, you only described how to build the alignment, but no explanation how to infer the tree.
6. Page 13, Internal nodes reconstruction: I have doubts about this section, the method used here sounds quite ad-hoc. Whereas there is already a full literature about model-based ancestral sequence reconstruction, such as empirical Bayesian method (e.g., <https://www.genetics.org/content/141/4/1641>). This method is available in widely used phylogenetics software. Therefore, I would use such method instead of the one used here.
7. Page 14, Substitution model: Here I'm confused about the terminology "substitution model". In phylogenetics this is something like GTR+I+G model for DNA data. However, this is not mentioned and I think you referred to something else. In that case please use another term. Moreover, you compute the AIC score but I have no idea how you compute the log-likelihood and degree of freedom of the model. Please explain.

Modeling SARS-CoV-2 substitution processes: predicting the next variant: Response letter

Keren Levinstein Hallak[†] Saharon Rosset^{*†}

November 21, 2021

We thank the reviewers for their thorough and detailed review. We have addressed all of the comments, a detailed description is given below.

Reviewer 1:

Major comments

1. *Some of the significances of the study may be overstated. For example, the authors said "Our methods can help vaccine design" in the Introduction (and similar statement in the Abstract), but the path to apply the methods and results from this study to vaccine design is still unclear to me – although some of the variants of interest are highly ranked (Table 3), there are still thousands of other candidates at similar ranks and it is challenging to pick up the few most important ones from them, and prediction of the prevalence of the variants in the population is a different topic.*

We agree that both points the reviewer raised require further clarification. To the first point, there is no doubt that mutations that are highly ranked in our list may not appear. To the second point, it is true that our method does not predict infectiousness or effectiveness as a drug/vaccine target, and certainly, it cannot predict prevalence which also depends on factors like government policy and population density. However, even given all these reservations, one can still see a clear path how an effective ranking algorithm like we propose can be useful in a vaccine or drug design pipeline, in conjunction with tools for predicting infectiousness like [1] and target effectiveness like [3].

All of these components can be beneficial for pipelines in the future: ranking by likelihood of occurrence (as we do), infectiousness, and treatability. While this does not pinpoint a specific variant to prepare for, going forward, it may still be practical to

*Corresponding author

[†]Department of Statistics and Operations Research, School of Mathematical Sciences, Tel-Aviv University, 6997801, Tel-Aviv, Israel

prepare many vaccine variants in advance (for example, with mRNA technology). So overall, we agree that it is appropriate to qualify the usefulness for vaccine design and have now added an explanation in the spirit above (last paragraph in the Discussion). However, we still believe in the potential usefulness of our results (and more generally, our statistical modeling approach) for drug and vaccine design.

2. *The authors said the testing data did not depend on the construction of the phylogenetic tree (page 11 lines 184 - 187), while the exposure was calculated based on the phylogenetic tree (page 15 lines 315 - 318). So how was the exposure determined for testing data? Did it implicitly depend on the constructed phylogenetic tree?*

The test data does not have exposure per-se since it does not have a tree. Since the test phase only used sites that have not mutated during the entire training period, the total amount of test “exposure” is by definition equal for all test sites, and there is no need to account for it in ranking by likelihood of mutations in the test. A slight complication is that the setting of the explanatory variables (and hence by implication the rate) may not be fixed for all training set leaves, which are “starting points” for test set analysis. In our analysis, the test exposure is implicitly divided between these rates according to training set leaves distribution. Under this assumption, there is still no need to include exposure in the testing phase explicitly. Critically, none of this violates the “purity” of the test analysis and hence the validity of the performance we obtain. We added a clarification to the revision (p. 15, line 351).

3. *As the SARS-CoV-2 virus is dynamically evolving, a relevant question is how the approach would perform for the new sequences and new amino acid changes emerging after April 10, 2021, further apart from the period during which the testing data was retrieved for the study (i.e. February 10, 2021 to April 10, 2021, which is immediately after the training cutoff date)?*

To address this, we have added a new test set of sequences collected between September 15th, 2021, and October 1st, 2021. We describe the details and results in the supplementary material. Overall, the results are very similar to those we got for the original test set, and confirm that they are not sensitive to the time of collection.

4. *How would the proposed approach perform as compared to existing methods? The authors said previous methods considered each predictor in isolation (page 2 line 34), so it may be informative to compare the top models with the base model + single factor one at a time in terms of AUC and lift.*

An important point here is that our methodology uses the training set for model selection and does not compare the performance of all models on the test set to preserve statistical validity. The models with only one explanatory factor (or models divided into sub-models by one explanatory factor) did not perform well in our training set

(ranks range from 15,417 to 21,570) and were therefore not evaluated on the test set. In order to address the reviewer's request, we address specific previous works according to which different genes have different mutation rates [4, 2], and CG pairs have a higher mutation rate [7, 6]. We evaluated the performance of the models divided into sub-models by genes and by the CG pair indicator (note, however, that these are models within our framework that incorporate our contributions like exposure, which improve results significantly compared to naive modeling). We describe here the results of these two models (model that divides by genes first, model that divides by CG pair indicator second): AUC scores of 0.825, 0.814, 3% lift scores versus the random model of 4.156, 2.654 and 3% lift scores versus the base model of 1.976, 1.262 for the non-synonymous amino acid substitutions Poisson model (results for the NB model are very similar). Compared to the ten top models, these results are substantially inferior (as shown in Table S4 in the supplementary).

We did not add these results to the manuscript to avoid confusing the reader, but we are happy to do so if the reviewer thinks it is warranted.

5. *What other sequence context relevant "internal" factors and other "external" factors beyond sequence relevant information could be possibly informative to predict variants of interest? This may be discussed in the Discussion section for future improvement of the prediction model as more data are accumulated.*

Thank you for the suggestions, we have added a paragraph to the Discussion in this spirit.

Minor comments

1. *On page 13, lines 272-273: "Finally, we applied a battery of statistical tests to validate the phylogenetic tree and its internal nodes." Although the authors gave an example, what are these statistical tests? Perhaps more details can be provided in supplementary information.*

We added details to the supplementary information.

2. *References 24 and 25 are the same.*

Fixed. Thanks for noticing!

Reviewer 2:

1. *The ms didn't make clear one selling point, that traditional phylogenetic methods can't predict future variants, although they're very good at finding and classifying existing variants based on the current data. This can be mentioned in the abstract and/or introduction.*

Thanks for the comment. We have added a clarification to the introduction (page 2, lines 37-39).

2. *Is there a reason why you only used the top-3 models but not top-10 or so?*

By presenting only the top-3 models, our goal was to ease the reader’s experience and emphasize that the test set should be used sparingly to preserve statistical integrity. To address the reviewer’s request we have added a description of the top-10 models to the supplementary, along with their AUC and lift results

3. *Page 9, Figure 2 caption: Do you mean “random (red) and base models (cyan)”? (colors swapped).*

Yes, sorry for the confusion. Thanks for noticing!

4. *Page 12, code availability, this is my main concern: The GitHub repo contains a collection of R scripts but absolutely no guide how to use them. Eventually I can’t do any analysis such as replicating the analysis done here. Readers will have the same problem if this is published. So please describe step-by-step usages of your scripts for reproducibility. Moreover, someone may want to use it to re-train the model on new data.*

We have added an elaborate read.me file with a description of the code’s workflow, input variables, and run instructions enabling to reproduce our results, re-train models, and test their predictions on new data. The code is available at:
<https://github.com/Kerenlh/sarscov2predictions.git>

5. *Page 12, Phylogeny of SARS-CoV-2: the authors used two unpublished methods from NCBI and sarscov2phylo github repo. That’s OK but you need to explain these two methods in more details. For example, you only described how to build the alignment, but no explanation how to infer the tree.*

Thanks for the comment. We added more details to the online methods section (p.13, lines 273-280).

6. *Page 13, Internal nodes reconstruction: I have doubts about this section, the method used here sounds quite ad-hoc. Whereas there is already a full literature about model-based ancestral sequence reconstruction, such as empirical Bayesian method (e.g., <https://www.genetics.org/content/141/4/1641>). This method is available in widely used phylogenetics software. Therefore, I would use such method instead of the one used here.*

We have searched the literature and consulted with an expert (Tal Pupko of Tel-Aviv University), and concluded that the mentioned methods and other leading model-based approaches (e.g. FastML [5], PAML [8]) cannot deal with the data sizes considered in our paper. It may be possible to design approximations using such model-based

approaches (say, by sampling subsets of sequences), but we feel that this is a major topic for research and is beyond the scope of the current paper.

We emphasize that our work inherently acknowledges that neither the phylogenetic reconstruction nor the ancestral sequence reconstruction of the training data are completely accurate. We account for that through the objective evaluation of the test data, that does not depend on such reconstructions. Thus, we prefer to keep the current ancestral sequence reconstruction approach. We added a clarification in the online methods section (p.14).

7. *Page 14, Substitution model: Here I'm confused about the terminology "substitution model". In phylogenetics this is something like GTR+I+G model for DNA data. However, this is not mentioned and I think you referred to something else. In that case please use another term. Moreover, you compute the AIC score but I have no idea how you compute the log-likelihood and degree of freedom of the model. Please explain.*

Thanks for this comment. Both issues are important, they are separate but related. First to the second point on AIC and log-likelihood: Our substitution models are composed of sub-models whose number is determined by the factors partitioning the model. Continuing the example mentioned in p. 15 line 326: A substitution model that contains the codon position as an explaining factor and the base as a partitioning factor will be composed of four sub-models, one for each base (A/C/G/U). Each sub-model will have $k = 3$ degrees of freedom for the Poisson regression (as there are three options for the codon position and an intercept) and $k = 4$ for the NB regression (another free parameter is required for θ in the NB regression). The log-likelihood is calculated for each sub-model regression separately and then summed up. The degrees of freedom are also summed up, allowing us to calculate the total AIC score of the model. We added a short clarification on this point in the paper (p. 15, lines 333-334).

Now to the first point on the substitution model: Since we are doing a Poisson regression (or similar but a little more involved NB regression), the resulting model can be thought of as describing a substitution rate for each possible mutation, as a function of not only the origin and target nucleotide, but also the setting of the other explanatory variables. Thus in the example above, we can think of it as three 4x4 rate matrices — one for each codon position.

References

- [1] Jiahui Chen, Rui Wang, Menglun Wang, and Guo-Wei Wei. Mutations strengthened sars-cov-2 infectivity. *Journal of molecular biology*, 432(19):5212–5226, 2020.
- [2] Maddalena Dilucca, Sergio Forcelloni, Alexandros G Georgakilas, Andrea Giansanti, and Athanasia Pavlopoulou. Codon usage and phenotypic divergences of sars-cov-2 genes. *Viruses*, 12(5):498, 2020.

- [3] David E Gordon, Gwendolyn M Jang, Mehdi Bouhaddou, Jiewei Xu, Kirsten Obernier, Kris M White, Matthew J O’Meara, Veronica V Rezelj, Jeffrey Z Guo, Danielle L Swaney, et al. A sars-cov-2 protein interaction map reveals targets for drug repurposing. *Nature*, 583(7816):459–468, 2020.
- [4] Neha Kaushal, Yogita Gupta, Mehendi Goyal, Svetlana F Khaiboullina, Manoj Baranwal, and Subhash C Verma. Mutational frequencies of sars-cov-2 genome during the beginning months of the outbreak in usa. *Pathogens*, 9(7):565, 2020.
- [5] Asher Moshe and Tal Pupko. Ancestral sequence reconstruction: accounting for structural information by averaging over replacement matrices. *Bioinformatics*, 35(15):2562–2568, 2019.
- [6] Mukhtar Sadykov, Tobias Mourier, Qingtian Guan, and Arnab Pain. Short sequence motif dynamics in the sars-cov-2 genome suggest a role for cytosine deamination in cpg reduction. *BioRxiv*, 2020.
- [7] Yong Wang, Jun-Ming Mao, Guang-Dong Wang, Zhi-Peng Luo, Liu Yang, Qin Yao, and Ke-Ping Chen. Human sars-cov-2 has evolved to reduce cg dinucleotide in its open reading frames. *Scientific Reports*, 10(1):1–10, 2020.
- [8] Ziheng Yang. Paml 4: phylogenetic analysis by maximum likelihood. *Molecular biology and evolution*, 24(8):1586–1591, 2007.

REVIEWERS' COMMENTS:

Reviewer #1 (Remarks to the Author):

The authors addressed all my questions, and I have no further comments.

Reviewer #2 (Remarks to the Author):

In this revision the authors have adequately addressed my concerns. I have no further comments.